# NExtLong: Toward Effective Long-Context Training without Long Documents

**Chaochen Gao** [1 2]  **Xing Wu** [1 2]  **Zijia Lin** [3]  **Debing Zhang** [4]  **Songlin Hu** [1 2]

## Abstract

Large language models (LLMs) with extended context windows have made significant strides yet remain a challenge due to the scarcity of long documents. Existing methods tend to synthesize long-context data but lack a clear mechanism to reinforce the long-range dependency modeling. To address this limitation, we propose NExtLong, a novel framework for synthesizing long-context data through Negative document Extension. NExtLong decomposes a document into multiple meta-chunks and extends the context by interleaving hard negative distractors retrieved from pretraining corpora. This approach compels the model to discriminate long-range dependent context from distracting content, enhancing its ability to model long-range dependencies. Extensive experiments demonstrate that NExtLong achieves significant performance improvements on the HELMET and RULER benchmarks compared to existing long-context synthesis approaches and leading models, which are trained on non-synthetic long documents. These findings highlight NExtLong's ability to reduce reliance on non-synthetic long documents, making it an effective framework for developing advanced long-context LLMs. Our code is available in https://github.com/caskcsg/longcontext/tree/main/NExtLong.

## 1. Introduction

Large language models (LLMs) have garnered significant attention due to their powerful and versatile capabilities. Recently, the context length of LLMs has been rapidly extended (Peng et al., 2023; AI et al., 2024; Yang et al., 2024;

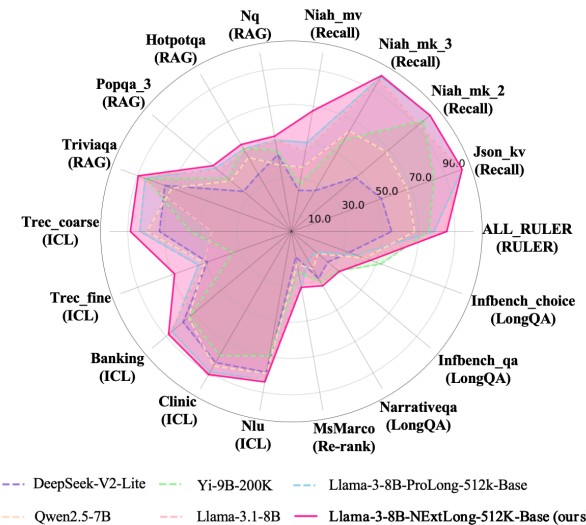

*Figure 1.* Comparison of existing remarkable models and NExtLong on the HELMET(Yen et al., 2024b) and RULER(Hsieh et al., 2024) benchmarks. We evaluate various task types classified by HELMET. All results are averaged over sequence lengths of 8K, 16K, 32K, 64K, and 128K.

Li et al., 2025). For example, the Llama series models increase the context length from 4k in Llama 2 (Touvron et al., 2023b) to 128K in Llama 3.1 (Meta, 2024). The increased context window enables LLM to unlock more challenging tasks, such as Document Summary (Wu et al., 2023b), Long-book QA (Caciularu et al., 2023) and Code Planning (Bairi et al., 2023). To model long-range dependencies, mainstream methods (Fu et al., 2024; Gao et al., 2024b) typically continue training existing LLMs pre-trained on a 4K or 8K context length with long documents that reach the target length, e.g., 128K. However, the scarcity of high-quality long documents [1] in most domains remains a significant challenge, particularly as the target context length continues to increase (Gao et al., 2024a).

To address the challenge of scarcity of long documents, existing approaches synthesize long-context data by concatenating shorter documents. Similarity-based methods, such

[1]Institute of Information Engineering, Chinese Academy of Sciences [2]School of Cyber Security, University of Chinese Academy of Sciences [3]Tsinghua University [4]Xiaohongshu Inc. Correspondence to: Xing Wu <wuxing@iie.ac.cn>, Songlin Hu <husonglin@iie.ac.cn>.

*Proceedings of the 42^{nd} International Conference on Machine Learning*, Vancouver, Canada. PMLR 267, 2025. Copyright 2025 by the author(s).

---

[1]In this work, we define "long-context data" as synthetic long-form datasets, and "long documents" as non-synthetic long documents that meet the target training length.

as KNN (Guu et al., 2020; Levine et al., 2021), aggregate the top-k semantically similar short documents into a longer document. Other studies (Roziere et al., 2023; Ouyang et al., 2022; Touvron et al., 2023a) randomly sample and concatenate short documents, often compromising coherence and relevance. Recently, Quest (Gao et al., 2024a) aims to balance semantic correlation and contextual diversity by retrieving documents relevant to specific keywords. However, those methods typically concatenate short documents based on random or similarity-based rankings, lacking a clear mechanism for capturing long-range dependencies.

An intuitive approach to building documents with long-range dependencies is to insert additional text between dependent segments (Tian et al., 2024; Zhao et al., 2024), thereby transforming short dependencies into long-range ones. However, recent studies show that large language models can be easily distracted by irrelevant context (Shi et al., 2023a), and this issue is exacerbated as the context length increases(Han et al., 2023). This raises a critical challenge: *how can we enhance a model's ability to discriminate long-range dependent information from distracting content within extended contexts?*

Inspired by the hard negative technique (Robinson et al., 2020; Kalantidis et al., 2020; Zhan et al., 2021) from contrastive learning, which introduces hard negatives to enhance a model's ability to discriminate relevant samples from distracting ones, we adapt this concept to create hard negative distractors that reinforce long-range dependency modeling. The key idea is to generate negative-extended documents by inserting semantically similar yet distracting texts between dependent fragments. These distractions increase the model's learning difficulty, thereby enhancing its capacity to model long-range dependencies. Specifically, NExtLong works by first chunking a document into multiple chunks, termed meta-chunks. We retrieve hard negatives from a pretraining corpus for meta-chunks and interleave them between dependent meta-chunks. Since the pretraining corpus undergoes extensive deduplication, these hard negatives share partial semantic similarities with the meta-chunks but do not replicate their content. By inserting these distractors between originally dependent meta-chunks, NExtLong not only increases the distance between dependent chunks—effectively transforming the dependencies into long-range ones—but also introduces distracting noise, which compels the model to reinforce its ability to discriminate long-range dependent context from distracting content.

Extensive experiments on the HELMET(Yen et al., 2024b) and RULER(Hsieh et al., 2024) benchmarks demonstrate that NExtLong significantly improves the model's ability to capture and utilize long-range dependencies. Overall, NExtLong delivers an average performance improvement of 7.33% over the previous long-context synthesis method

Quest (Gao et al., 2024a). Moreover, compared to existing remarkable models trained by long documents, NExtLong achieves extraordinary results, as highlighted in Figure 1. These results demonstrate that NExtLong is a highly effective method for synthesizing long-context data. The synthesized data significantly alleviates the dependence on training large long-context models on long documents and holds the potential to train ultra-long context models that are not constrained by the scarcity of long documents.

Our main contributions can be summarized as follows:

- We propose NExtLong, a simple and effective method that extends the document to strengthen the model's ability to model long-range dependencies.

- We provide an in-depth analysis of the key components that contribute to the effectiveness of NExtLong.

- Our experiments show that NExtLong achieves a significant improvement across multiple long-context evaluation tasks, demonstrating the effectiveness of negative document extension in training long-context LLMs.

## 2. Related Work

**Unlocking LLMs' ability to process long-context tasks. Train-free methods** bypass parameter updates for long-context handling. LM-Infinite (Han et al., 2023) employs a $\Lambda$-shaped attention mask with a distance limit for length generalization. StreamingLLM (Xiao et al., 2023) mitigates the "attention sink" phenomenon by balancing attention scores. Self-Extend (Jin et al., 2024b) introduces group-wise attention to map unseen relative positions, while DCA (An et al., 2024) uses token-wise attention and memory-efficient mechanisms for effective context extension. **Train-based methods** enhance performance through continued training. Chen et al. (2023b) extend RoPE-based (Su et al., 2021) LLMs via positional interpolation, and PoSE (Zhu et al., 2023) applies positional skip-wise training to decouple training and target lengths. Recently, upsampling long documents across diverse domains has emerged as a critical factor in advancing long-context modeling (Fu et al., 2024; Gao et al., 2024b; Xiong et al., 2023).

However, those train-based methods are dependent on the availability of high-quality long documents, which are scarce in many domains and become increasingly harder to obtain as context lengths grow. In contrast, NExtLong overcomes this challenge by extending documents with hard negatives, alleviating the reliance on naturally occurring long documents.

**Synthesizing long-context texts by concatenating short documents.** Past approaches to synthesizing long-context data primarily rely on concatenating short documents.

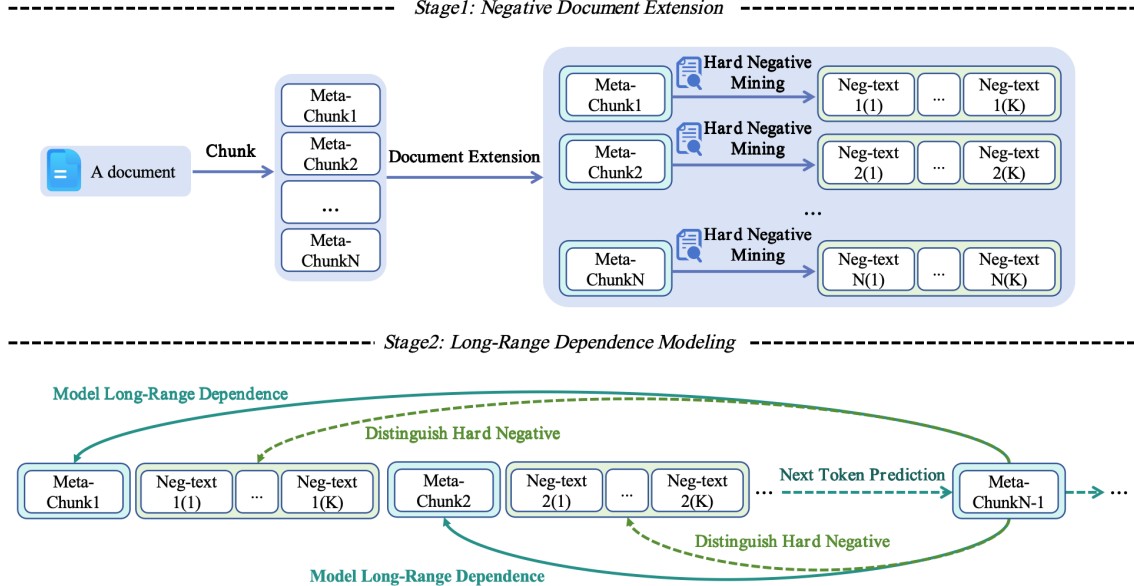

*Figure 2.* The NExtLong method consists of two stages. In the first stage, a document is chunked into multiple meta-chunks, and each meta-chunk is mined for numerous hard negatives. These hard negatives are then concatenated with the meta-chunks to create a long document. In the second stage, the model is trained using this synthesized long document, focusing on modeling long-range dependencies by identifying the meta-chunks across a wide range of hard negatives.

Those methods often lack a mechanism to ensure that the concatenated documents maintain explicit long-range dependencies. Some methods randomly sample and concatenate short documents (Roziere et al., 2023; Chen et al., 2023c), while others attempt to preserve semantic coherence by clustering similar documents using KNN (Guu et al., 2020). Recent works like Quest (Gao et al., 2024a) try to balance semantic relevance and diversity by retrieving keyword-related documents. However, those methods typically fail to hold a mechanism to effectively model long-range dependencies across distant documents. In contrast, NExtLong not only synthesizes long-context data but also explicitly introduces hard negatives between document chunks, which reinforces the model's ability to learn and utilize long-range dependencies.

**Hard negative technique.** Hard negative mining is a well-established technique in contrastive learning and dense retrieval, aimed at improving model discrimination by introducing samples that are semantically similar yet incorrect (Robinson et al., 2020; Xiong et al., 2020). In dense retrieval, it enables the model to effectively distinguish between relevant and irrelevant information by selecting challenging negative samples (Zhan et al., 2021; Wu et al., 2023a). While traditionally applied to retrieval tasks, recent studies explore its potential in LLMs, such as Jin et al. (2024a), which investigates the role of hard negatives in Retrieval-Augmented Generation (RAG). However, those approaches have not yet been applied to document synthesis for adapting LLMs to

handle longer contexts. In this work, we adapt hard negative mining for document synthesis by interleaving hard negatives between meta-chunks in NExtLong, which helps the model better focus on long-range dependent context and improves its ability to process long-context tasks.

## 3. Method

This section introduces our proposed method, NExtLong, which comprises two stages: **Negative Document Extension** and **Long-Range Dependence Modeling**. NExtLong aims to enhance long-context modeling by synthesizing extended-length documents. An overview of the approach is shown in Figure 2, and the corresponding pseudocode is presented in Appendix D.2.

### 3.1. Negative Document Extension

The Negative Document Extension stage consists of two steps: document chunking and hard negative mining.

#### 3.1.1. DOCUMENT CHUNKING

We sample a document from the training dataset as a meta-document $r$ and divide it into sequential meta-chunks. We define the documents to be expanded as meta-documents. The meta-document is divided into several chunks according to a certain chunking granularity. These chunks are defined as meta-chunks. To ensure sentence integrity, we define a maximum length $s$ as the chunking granularity. The

chunking process follows a two-step approach:

1. **Splitted by newline:** The meta-document $r$ is first splitted into paragraphs based on newline characters (\n), preserving the coherence of each paragraph.

2. **Form chunks:** These paragraphs are concatenated sequentially to form meta-chunks $m_i$ until the cumulative length reaches the maximum length $s$. If adding another paragraph exceeds this threshold, the current group is finalized as a complete meta-chunk, and the process continues with the remaining text. The effect of chunking granularity $s$ is analyzed in Section 5.3.

In this way, the meta-document $r$ is divided into $p$ meta-chunks:

$$r \xrightarrow{\text{chunk}} \{m_1, m_2, \ldots, m_p\} \tag{1}$$

The number of meta-chunks $p$ depends on the length of the meta-document $r$ and the chunking granularity $s$.

### 3.1.2. HARD NEGATIVE MINING

To obtain distracting texts as hard negatives for each meta-chunk, we build a FAISS index from the pretraining dataset, which undergoes extensive deduplication. Unlike methods that treat entire documents as indivisible units, we also divide each document in the pretraining dataset into smaller chunks based on the same granularity $s$. This chunking enables more fine-grained and efficient content retrieval during the extension process. Formally, each document $d_i$ in the pretraining corpus is divided into $q$ chunks:

$$d_i \xrightarrow{\text{chunk}} \left\{c_{i_1}, c_{i_2}, \ldots, c_{i_q}\right\} \tag{2}$$

Each chunk is indexed individually for precise and efficient retrieval. We compute the embedding vector $e_i$ for each chunk and insert it into the FAISS index:

$$\left\{c_{i_1}, c_{i_2}, \ldots, c_{i_q}\right\} \xrightarrow{\text{project}} \left\{e_{i_1}, e_{i_2}, \ldots, e_{i_q}\right\} \xrightarrow{\text{index}} \text{FAISS} \tag{3}$$

After building the FAISS index, we retrieve the top-$k$ most similar chunks as hard negatives $n_{i_j}$ for each meta-chunk $m_i$. These hard negatives are then concatenated with the meta-chunk to form an extended chunk $l_i$:

$$l_i = [m_i, n_{i_1}, n_{i_2}, \ldots, n_{i_k}] \tag{4}$$

We conduct ablation experiments on the position of meta-chunks (Appendix B.1), confirming that placing the meta-chunk before the hard negatives yields better performance. The number of hard negatives, i.e., $k$, depends on the length of the meta-document, chunking granularity $s$, and target

context length. Details for calculating $k$ are provided in Appendix D.1.

Finally, we synthesize a long document $t$ by concatenating the extended chunks:

$$t = [l_1, l_2, \ldots, l_p] \tag{5}$$

### 3.2. Long-Range Dependence Modeling

In alignment with the pretraining stage, we employ next token prediction (NTP) loss (Radford, 2018) to extend the context length of the base model. The loss function is defined as:

$$\text{Loss} = -\sum_{t=1}^{T} \log P(x_{t+1}|x_1, x_2, \ldots, x_t) \tag{6}$$

The key distinction of NExtLong lies in the differentiation of tokens during training. The tokens in the synthesized long document $t$ are classified into two categories: meta-chunks $m_i$ and hard negatives $n_{i_j}$. Together, they form an extended chunk $l_i$. For simplicity, we use $m_i$, $n_{i_j}$, and $l_i$ to denote the encoded tokens of meta-chunks, hard negatives, and extended chunks, respectively. The loss function can thus be reformulated as:

$$\text{Loss} = -\sum_{t=1}^{T} \log P(x_{t+1}|m_1, n_{1_1}, n_{1_2}, \ldots, x_t)$$
$$= -\sum_{t=1}^{T} \log P(x_{t+1}|l_1, l_2, \ldots, x_t) \tag{7}$$

The NTP loss encourages the model to distinguish relevant meta-chunks from surrounding hard negatives and to model long-range dependencies effectively. In Section 4 and Section 5, we empirically demonstrate that incorporating hard negatives in the loss function improves the model's ability to model long-range dependencies across extensive contexts.

## 4. Experiments

In this section, we evaluate the effectiveness of NExtLong by comparing it with other data synthesis methods (Section 4.2) and state-of-the-art (SOTA) models (Section 4.3).

### 4.1. Experimental Setups

**Datasets** We select two commonly used pretraining datasets composed entirely of short documents (Refer to Appendix B.2 for document length distribution): Cosmopedia v2 (Ben Allal et al., 2024) and FineWeb-Edu (Lozhkov et al., 2024). Both datasets are used for the main experi-

*Table 1.* Comparing NExtLong with other data synthesis methods on HELMET and RULER benchmark. All results are averaged over sequence lengths of 8K,16K,32K,64K, and 128K. ◇: results from Yen et al. (2024b);♣: results evaluated by ourselves.

| Model | Max Len | Avg. | Recall | RAG | ICL | Re-rank | LongQA | RULER |
|---|---|---|---|---|---|---|---|---|
| Meta-Llama-3-8B-base ◇ | 8K | 13.37 | 18.00 | 12.68 | 13.60 | 7.74 | 10.38 | 17.80 |
| + *Standard* ♣ | 128K | 52.85 | 62.33 | 58.67 | 71.24 | 19.18 | 28.99 | 76.68 |
| + KNN ♣ | 128K | 50.97 | 64.24 | 56.00 | 60.28 | 18.77 | 32.27 | 74.30 |
| + ICLM ♣ | 128K | 50.37 | 64.04 | 54.48 | 72.36 | 14.04 | 28.17 | 69.14 |
| + Quest ♣ | 128K | 55.25 | 69.13 | 57.47 | 72.08 | 22.35 | 33.82 | 76.63 |
| + NExtLong (ours) ♣ | 128K | **62.58** | **82.56** | **60.91** | **81.76** | **31.47** | **37.30** | **81.50** |

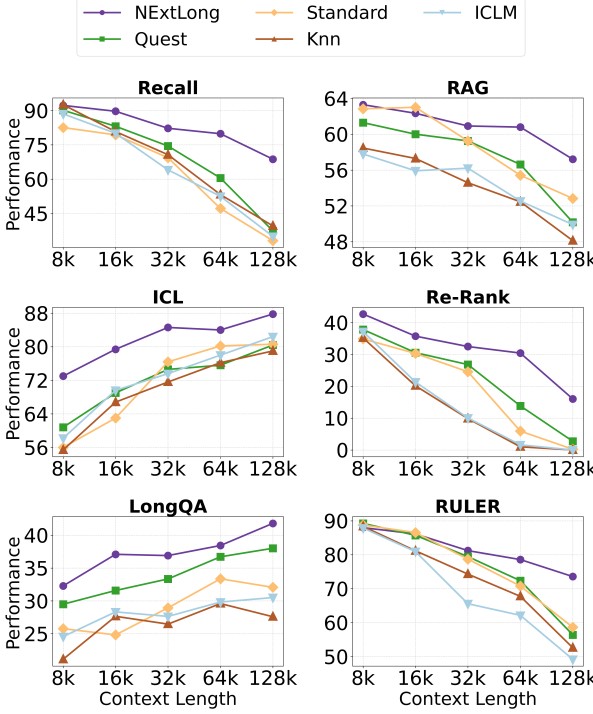

*Figure 3.* Comparison of NExtLong with other data synthesis methods on HELMET and RULER benchmarks across different context lengths. NExtLong shows significant performance improvements across various tasks.

ments, and we also provide ablation studies on their selection in Appendix B.3. Various methods, including NExtLong and baseline approaches, are employed to synthesize target-length samples concatenated from these short documents. The datasets are described as follows:

- **Cosmopedia v2:** An advanced version of the largest synthetic dataset for pretraining, comprising over 39 million generated samples from textbooks, blog posts, and stories.
- **FineWeb-Edu:** Consists of 1.3 trillion tokens of educational web pages filtered from the FineWeb dataset.

**Evaluation** Recent long-context evaluations have focused on a 128K context length (Zhang et al., 2024c; Hsieh et al.,

2024; Yen et al., 2024b), leading to the creation of various evaluation datasets. Accordingly, we set the target context length to 128K for comprehensive evaluation. We evaluate the models using the HELMET (Yen et al., 2024b) and RULER (Hsieh et al., 2024) benchmarks. The evaluation spans five task types from the HELMET benchmark: synthetic recall, retrieval-augmented generation (RAG), many-shot in-context learning (ICL), passage re-ranking, and long-document QA, covering a total of 17 sub-tasks. Detailed descriptions of the HELMET benchmarks can be found in Appendix C.3. Additionally, the RULER benchmark includes 13 synthesis sub-tasks.

### 4.2. Comparison with Other Data Synthesis Methods

We first compare NExtLong with previous long-context data synthesis methods on the 128K context length setting.

**Experimental Settings for Extending Context Length to 128K.** We fine-tune the Meta-Llama-3-8B-base (Meta, 2024) model using a batch size of 4M tokens for 1000 steps with the open-source framework GPT-NeoX[2]. The RoPE frequency base is increased from 500,000 in Meta-Llama-3-8B-base to 200,000,000. The same training configuration is applied to all methods for a fair comparison. Further details are available in Appendix C.1.

**Baseline Methods** We compare NExtLong with several methods that synthesized 32,000 128K-length samples (approximately 4 billion training tokens) from short documents:

- **Standard Method:** Randomly samples and concatenates short documents (Ouyang et al., 2022; Le Scao et al., 2023; Touvron et al., 2023a).
- **KNN (Guu et al., 2020; Levine et al., 2021):** Pairs each document with the top $k$ most similar retrieved documents.
- **ICLM (Shi et al., 2023b):** Uses a traveling salesman algorithm to reduce redundancy and improve diversity.
- **Quest (Gao et al., 2024a):** Balances semantic correlation and diversity by clustering documents based on predicted queries.

[2]https://github.com/EleutherAI/gpt-neox

*Table 2.* Comparing NExtLong with other open-source base models on the HELMET and RULER benchmarks. All results are averaged over sequence lengths of 8K, 16K, 32K, 64K, and 128K. ◇: results from Yen et al. (2024b); ♣: results evaluated by ourselves.

| Model | Max Len | Avg. | Recall | RAG | ICL | Re-rank | LongQA | RULER |
|---|---|---|---|---|---|---|---|---|
| *Open-source base models* | | | | | | | | |
| Yarn-Llama-2-7b-128K ♣ | 128K | 35.61 | 18.58 | 43.47 | 71.32 | 13.27 | 25.91 | 41.14 |
| DeepSeek-V2-Lite ♣ | 160K | 42.62 | 37.00 | 46.93 | 72.36 | 14.31 | 29.97 | 55.17 |
| Yi-9B-200K ♣ | 200K | 53.91 | 65.88 | 57.31 | 62.36 | 22.86 | **39.47** | 75.61 |
| Qwen2.5-7B ♣ | 128K | 49.53 | 59.04 | 49.44 | 73.84 | 18.37 | 28.62 | 67.84 |
| Mistral-Nemo-Base ♣ | 128K | 50.34 | 57.04 | 54.73 | 74.68 | 18.98 | 35.53 | 61.08 |
| Llama-3-8B-ProLong-512K-Base ♣ | 512K | 60.34 | 86.95 | 60.93 | 79.20 | **31.66** | 24.68 | 78.60 |
| Llama-3.1-8B ♣ | 128K | 61.07 | 83.18 | 61.22 | 71.40 | 29.10 | 37.35 | 84.20 |
| Llama-3-8B-NExtLong-512K-Base (ours) ♣ | 512K | **65.76** | **91.58** | **63.68** | **84.08** | 31.27 | 38.42 | **85.52** |
| *Closed-source models* | | | | | | | | |
| GPT-4o-mini ◇ | 128K | 70.98 | 94.90 | 69.64 | 78.12 | 52.30 | 43.13 | 87.82 |
| GPT-4o ◇ | 128K | 77.43 | 98.22 | 72.30 | 85.76 | 64.70 | 47.61 | 95.96 |
| Gemini-1.5-Pro ◇ | 2M | 71.71 | 84.60 | 72.06 | 78.74 | 69.04 | 45.99 | 79.84 |
| Claude-3.5-sonnet ◇ | 200K | 51.19 | 93.30 | 41.10 | 59.80 | 9.10 | 10.83 | 93.00 |

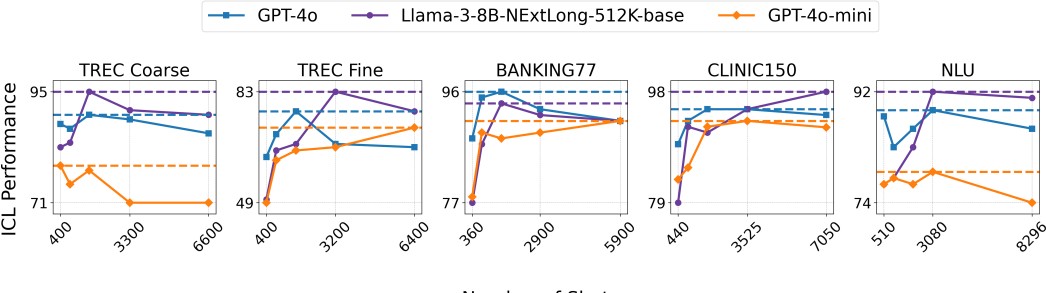

*Figure 4.* Comparison of NExtLong with GPT-4o on five In-Context Learning (ICL) tasks from the HELMET benchmark. Each polyline represents the model's performance across context lengths of 8K, 16K, 32K, 64K, and 128K.

**NExtLong Outperforms Existing Data Synthesis Methods.** Table 1 and Figure 3 compare NExtLong with other data synthesis methods across different context lengths (8K, 16K, 32K, 64K, and 128K) on the HELMET and RULER benchmarks. Table 1 presents the averaged results, indicating that NExtLong surpasses all baseline methods with an average improvement of at least +7.33%. Notably, it achieves a 13.43% gain in Recall and a 9.12% improvement in Re-Rank over the Quest method, demonstrating its effectiveness in enhancing long-context performance.

Figure 3 further illustrates that NExtLong outperforms other methods across varying context lengths, with the gap widening as context length increases. The results highlight NExtLong's superior capability to model long-range dependencies and maintain robust performance even at 128K context length, demonstrating the versatility and reliability of NExtLong across diverse tasks and its effectiveness in handling ultra-long contexts.

### 4.3. Comparison with SOTA Models

We compare NExtLong-trained models with state-of-the-art models, including ProLong (Gao et al., 2024b), which uses a two-stage training strategy: first training on shorter contexts, then extending to longer ones (e.g., 512K). ProLong evaluates models using a "train-long, test-short" approach, testing on shorter contexts (e.g., 128K). For fairness, we adopt the same strategy, training up to 512K and evaluating on 128K benchmarks.

**Experimental Settings for Extending Context Length to 512K.** Unlike other models such as ProLong (Gao et al., 2024b), which are trained on naturally occurring long documents, we utilize NExtLong-synthesized data for training. Specifically, we synthesize two long-context datasets, NExtLong-64K and NExtLong-512K, both derived from the FineWeb-Edu and Cosmopedia v2 corpora. The detailed training hyper-parameters are provided in Appendix C.2.

**Baseline Models** We select open-source base models with comparable parameter sizes for evaluation, including Llama-3.1-8B and Llama-3-8B-ProLong-512K-Base. Additionally, we compare against current SOTA closed-source models, such as GPT-4o, Gemini, and Claude.

*Table 3.* Comparing NExtLong with ProLong models on the LongBench v2 benchmark.

| Model | Overall | Easy | Hard | Short | Medium | Long |
|---|---|---|---|---|---|---|
| Llama-3-8B-ProLong-512K-Instruct | 27.2 | 31.8 | 24.4 | 31.7 | 29.3 | 15.7 |
| Llama-3-8B-NExtLong-512K-Instruct | **30.4** | **33.3** | **28.6** | **32.2** | **30.7** | **26.9** |

**Without Using Long Documents, NExtLong Outperforms Other Open-Source Models.** Table 2 shows that Llama-3-8B-NExtLong-512K-Base model surpasses other open-source models, outperforming Llama-3-8B-ProLong-512K-Base by +5.42% and Llama-3.1-8B by +4.69% on average. These results demonstrate that synthesized data can match or even surpass non-synthesized long documents in enhancing long-context capabilities, positioning NExtLong for ultralong context extensions.

**In ICL Tasks, NExtLong Matches or Surpasses GPT-4o as the Number of Shots Increases.** Recently, Long-context models' ICL performance has garnered significant attention (Bertsch et al., 2024; Agarwal et al., 2024; Anil et al., 2024). Agarwal et al. (2024) highlight ICL performance as a valuable metric for evaluating long-context models. Figure 4 shows that as the number of shots increases, NExtLong matches GPT-4o in the Banking77 task and outperforms it in four other tasks. Its strong performance and moderate computational cost make NExtLong suitable for future ICL applications.

## 5. Analysis

This section provides an in-depth analysis of the NExtLong method. Due to the high computational cost of experiments, ablation studies are conducted with a 128K context length.

### 5.1. NExtLong Enhances Long-Range Dependency Modeling

To assess the improvement in long-range dependency modeling achieved by NExtLong's negative document extension, we conduct a probing experiment using the Longbook QA dataset (Zhang et al., 2024b), which features long-range dependencies up to 128K in length. In this experiment, we use the normalized attention weights assigned to the first third of the context, when predicting the last token, as a metric for evaluating the model's long-dependency modeling ability.

As shown in Figure 5, we observe a positive correlation between this long-dependency metric and the model's performance on LongQA. Complementarily, as discussed in Appendix B.4, NExtLong reduces the model's dependence on proximal text (the last third context). These findings demonstrate that models trained with NExtLong's negative document extension exhibit enhanced long-dependency modeling capabilities, resulting in significantly improved long-context performance.

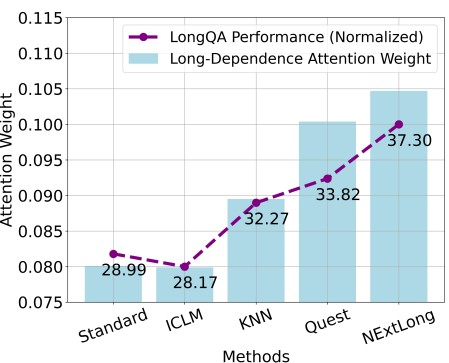

*Figure 5.* NExtLong enhances long-range dependency modeling. The bars represent the model's ability to capture long-range dependencies, measured by the attention weights assigned to the first third of the context. The dotted line indicates the model's performance, demonstrating a positive correlation between improved long-range dependency modeling and better performance on the LongQA task.

### 5.2. NExtLong Performs Strongly After Supervised Finetuning.

To evaluate how NExtLong performs after supervised fine-tuning, we follow the approach in ProLong (Gao et al., 2024b) and fine-tune our base model using the UltraChat (Ding et al., 2023) short-context SFT dataset. We test the model on the recently proposed LongBench v2 benchmark (Bai et al., 2024). As shown in Table 3, NExtLong outperforms ProLong overall, especially on the Long metric. The results demonstrate that Llama-3-8B-NExtLong-512K-Base performs strongly as a base model. With the same SFT dataset, the improved long-context base model enables the training of a superior fine-tuned model.

### 5.3. The Impact of Chunking Granularity $s$

We perform an ablation study on chunking granularity $s$ using values of 512, 1024, 2048, 8192, and 32768. The results, shown in Figure 6, indicate that the model performs best with a granularity of 2048. While a granularity of 1024 yields optimal performance for 128K context length, it underperforms in the 8k and 16k ranges compared to 2048. We conclude that too small a granularity disrupts semantic integrity, while too large introduces redundant information, negatively impacting the hard negative mining stage. A moderate granularity offers the best balance for performance.

*Table 4.* Comparison of short text performance across methods. Overall, NExtLong shows a minor performance fluctuation in short text benchmark as the method improves the long-context ability of Meta-Llama-3-8B-base.

| Model | Avg. | Hel. | Lam. | AR-C. | AR-E. | PIQA | Win. | Log. |
|---|---|---|---|---|---|---|---|---|
| Meta-Llama-3-8B-base | 63.75 | 60.13 | 75.66 | 50.34 | 80.18 | 79.60 | 72.85 | 27.50 |
| + *Standard* | 63.66 | 59.57 | 72.87 | 49.83 | 81.73 | 80.36 | 73.64 | 27.65 |
| + KNN | 63.23 | 59.74 | 72.25 | 49.15 | 80.85 | 80.30 | 73.56 | 26.73 |
| + ICLM | 63.89 | 61.24 | 71.67 | 52.47 | 81.73 | 80.14 | 73.56 | 26.42 |
| + Quest | 63.96 | 59.72 | 73.20 | 50.68 | 81.14 | 80.41 | 74.74 | 27.80 |
| + NExtLong | 63.83 | 60.32 | 72.06 | 51.45 | 82.03 | 79.87 | 73.09 | 27.96 |

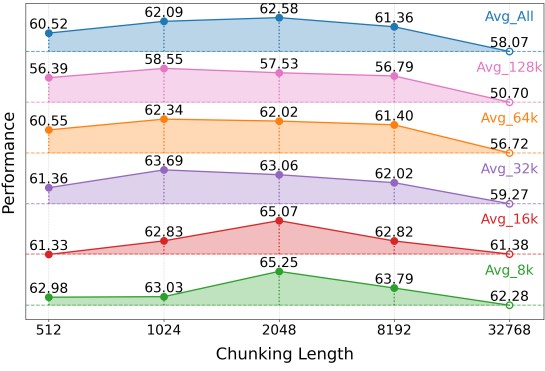

*Figure 6.* The impact of different chunking granularities $S$ on the performance of NExtLong. The six curves, from bottom to top, correspond to the average performance across six task types at document lengths of 8K, 16K, 32K, 64K, and 128K, as well as the overall average across all these lengths.

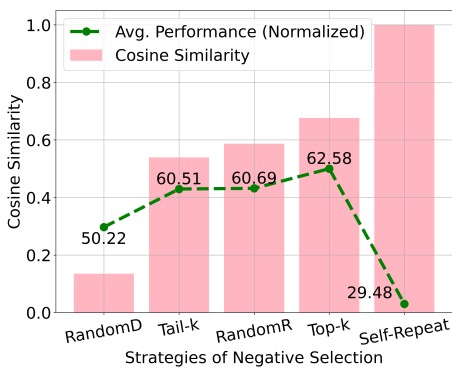

*Figure 7.* The impact of negative selection on long-context performance. The bars represent the cosine similarity of documents concatenated by different strategies. The dotted line indicates the average performance on the HELMET and RULER benchmarks, with all results normalized to align within the specified similarity range.

### 5.4. The Importance of Hard Negatives for Achieving Better Results

To evaluate the impact of hard negatives on performance, we design 5 document retrieval strategies. For each meta-chunk, we retrieve 512 documents from the Faiss index and select $k$ from these documents using the following strategies:

1. **Self-Repeat**: Repeat meta-chunks without including retrieved documents.

2. **Top-k (hard negatives)**: Concatenate documents in descending order of similarity until the target length is reached.

3. **RandomR**: Shuffle retrieved documents randomly and select from them.

4. **Tail-k**: Concatenate documents in ascending order of similarity.

5. **RandomD**: Randomly select documents from the training dataset, ignoring retrieval documents, which share a similar idea to (Tian et al., 2024).

Figure 7 shows that the choice of hard negatives (the **top-k** setting) plays a crucial role in NExtLong. Low-similarity negatives reduce training difficulty, weakening performance.

Meanwhile, using repeated meta-chunks brings false negatives, further degrades model performance, which is consistent with the phenomenon observed in contrastive learning (Chen et al., 2021).

### 5.5. NExtLong Shows No Significant Performance Degradation on Short Text.

To verify how well NExtLong maintains model performance on short text tasks, following Quest (Gao et al., 2024a), we select 7 widely-used short-text datasets: HellaSwag (Hel.) (Zellers et al., 2019), Lambada_OpenAI (Lam.) (Paperno et al., 2016), ARC-Challenge (AR-C.) (Clark et al., 2018), ARC-Easy (AR-E.), PIQA (Bisk et al., 2020), WinoGrande (Win.) (Sakaguchi et al., 2021), and Logiqa (Log.) (Liu et al., 2020). As shown in Table 4, compared to the Meta-Llama-3-8B-base model, the performance on short text evaluations shows no significant degradation after continued training with long-context data synthesized by NExtLong.

## 6. Conclusion and Future Works

This paper introduces **NExtLong**, a framework that improves long-range dependency modeling in LLMs through

negative document extension. By dividing a document into meta-chunks and inserting hard negative distractors, NExtLong increases learning difficulty, encouraging the LLMs to better model long-range dependencies over extended contexts. Experimental results show that NExtLong outperforms existing methods on HELMET and RULER benchmarks, achieving notable performance gains. In the future, we plan to explore more effective negative chunk mining strategies, such as generative approaches to creating more diverse and harder distractors, further enhancing the model's ability to learn fine-grained long-range dependencies.

## Acknowledgement

This work is supported by the National Natural Science Foundation of China (No. U24A20335).

## Impact Statement

This paper presents work whose goal is to advance the field of Machine Learning. There are many potential societal consequences of our work, none which we feel must be specifically highlighted here.

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

# A. The Result on Needle-in-a-Haystack Benchmark

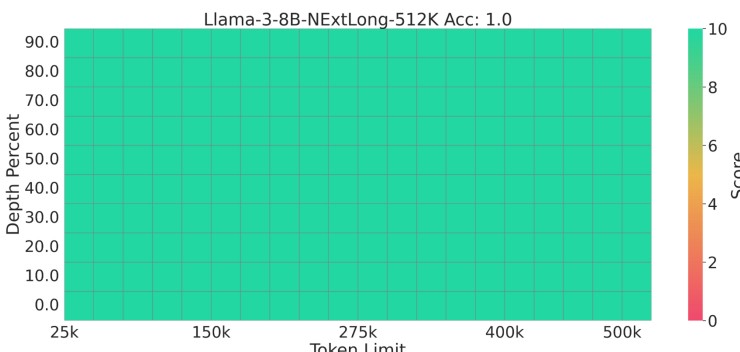

*Figure 8.* The Needle-in-a-Haystack task assesses a model's capability to extract specific information (the needle) from a large corpus of documents (the haystack). The y-axis indicates the position of the "needle" within the document, spanning from 25K to 500K tokens.

Following previous works (Gao et al., 2024a; Zhang et al., 2024a; Liu et al., 2024), we evaluate the Llama-3-8B-NExtLong-512K-Base model on the widely used Needle-in-a-Haystack task. As shown in Figure 8, Llama-3-8B-NExtLong-512K-Base achieves a 100% accuracy on the Needle-in-a-Haystack task.

# B. More Ablations

### B.1. Placing Meta-Chunk at Different Positions

We explore three different strategies for combining meta-chunk and hard negatives, which are represented by the following descriptions:

1. **Head**: Placing the meta-chunk at the beginning of the retrieved hard negatives.
2. **Tail**: Placing the meta-chunk at the end of the retrieved hard negatives.
3. **Random**: Randomly inserting the meta-chunk within the retrieved hard negatives.

Table 5 shows that placing the meta-chunk at the beginning (Head) yields better performance. We believe this method helps establish longer dependencies, resulting in enhanced effectiveness.

*Table 5.* Performance Comparison of Different Insertion Strategies. All results are averaged over sequence lengths of 8K,16K,32K,64K, and 128K.

| Model | Avg. | Recall | RAG | ICL | Re-rank | LongQA | RULER |
|---|---|---|---|---|---|---|---|
| Head | **62.58** | **82.56** | 60.91 | **81.76** | 31.47 | **37.30** | **81.50** |
| Tail | 60.01 | 72.66 | **63.67** | 80.68 | 30.73 | 34.09 | 78.26 |
| Random | 58.95 | 74.51 | 63.05 | 71.64 | **32.78** | 33.00 | 78.73 |

### B.2. Document Length Distribution of Cosmopedia V2 and FineWebEdu

We analyze the document length distribution of two datasets, Cosmopedia V2 and FineWebEdu, by sampling 8 million documents from each dataset and encoding them using the Meta-Llama-3-8B tokenizer. Document lengths are categorized into two ranges: $[0, 8192]$ and $> 8192$. Table 6 shows that the majority of documents in both datasets are relatively short (under 8K). We apply the NExtLong algorithm to extend the document length to 128K and 512K, achieving approximately a 64-fold increase compared to the original.

*Table 6.* Document Length Distribution of Cosmopedia V2 and FineWebEdu.

| Dataset | $0 \leq$ Length $\leq 8192$ | Length $> 8192$ |
|---|---|---|
| Cosmopedia V2 | 100.00% | 0.00% |
| FineWebEdu | 99.19% | 0.81% |

### B.3. Dataset Ablation Study

We compared three different dataset selection strategies: (1) using FineWeb-Edu alone for long-context data synthesis, (2) using Cosmopedia v2 alone for long-context data synthesis, and (3) combining both datasets for long-context data synthesis. The results are shown in Table 7. The findings indicate that the combined strategy achieved the best performance, highlighting that a diverse dataset significantly enhances data synthesis.

*Table 7.* Dataset Comparison on HELMET and RULER Benchmark.

| Dataset | Avg. | Recall | RAG | ICL | Re-rank | LongQA | RULER |
|---|---|---|---|---|---|---|---|
| Cosmopedia | 59.26 | 79.94 | 59.58 | 81.00 | 27.07 | 29.13 | 78.84 |
| FineWebEdu | 62.04 | **83.56** | **61.53** | **83.76** | 28.31 | 34.46 | 80.60 |
| Cosmopedia + FineWebEdu | **62.58** | 82.56 | 60.91 | 81.76 | **31.47** | **37.30** | **81.50** |

### B.4. NExtLong Reduces Dependence on Proximal Text

Complementary with Section 5.1, we investigated the dependence of different models on proximal text (the last third of the text). As shown in Figure 9, NExtLong demonstrates a lower degree of dependence on proximal text. This shift in attention toward long-range text contributes to improving the model's performance.

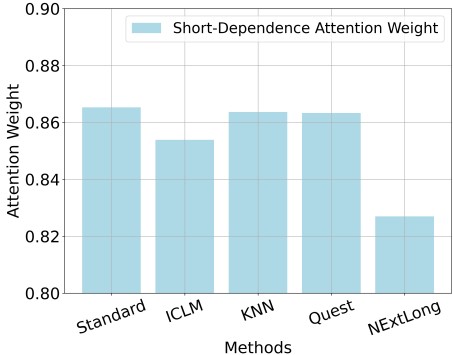

*Figure 9.* NExtLong reduces the model's dependence on proximal text. We calculated the degree of dependence on proximal text (the last third of the text) for different methods when completing the LongQA task. It can be observed that NExtLong significantly reduces the model's dependence on proximal text.

## C. Experiment Details

### C.1. Training Llama-3-8B-NExtLong-128K detailed setup

We use the parameters listed in Table 8 to train the 128K model. For other data synthesis methods, we only modify the training dataset while keeping all other training parameters unchanged.

### C.2. Training Llama-3-8B-NExtLong-512K-Base detailed setup

Given that mixing documents of different lengths during training is widely adopted in prior work (Gao et al., 2024b; Meta, 2024), we follow the ProLong (Gao et al., 2024b) approach by synthesizing both 512K-length and 64K-length documents for training the 512K NExtLong model. For synthetic 512K-length documents, we apply the full attention mechanism, as each document constitutes an entire training sample. For synthetic 64K-length documents, we concatenate them into 512K-length training samples (each sample contains eight 64K-length documents) and employ intra-document attention (Ding et al., 2024) to restrict information flow within each 64K-length document. We use the parameters listed in Table 9 to train the 512K model. The training samples are sourced from the NExtLong-512K and NExtLong-64k datasets in a ratio of 1:2.

*Table 8.* 128K model training configuration.

| 128K training setting | |
|---|---|
| Initial Model | Meta-Llama-3-8B (base model) |
| rotary-emb-base | 200,000,000 |
| $\beta_1$ | 0.9 |
| $\beta_2$ | 0.95 |
| lr | $4e^{-5}$ |
| precision | bfloat16 |
| gradient-clipping | 1.0 |
| weight-decay | 0.1 |
| lr-decay-style | cosine |
| train-iters | 1000 |
| warmup-iters | 200 |
| seq-length | 131072 |
| GPU-type | H100 |
| GPU-numbers | 64 |
| training-time | 15h |

*Table 9.* 512K model training configuration.

| 512K training setting | |
|---|---|
| Initial Model | Llama-3-8B-ProLong-64k-Base |
| rotary-emb-base | 128,000,000 |
| $\beta_1$ | 0.9 |
| $\beta_2$ | 0.95 |
| lr | $1e^{-5}$ |
| precision | bfloat16 |
| gradient-clipping | 1.0 |
| weight-decay | 0.1 |
| lr-decay-style | cosine |
| train-iters | 500 |
| warmup-iters | 50 |
| seq-length | 524288 |
| GPU-type | H100 |
| GPU-numbers | 128 |
| training-time | 20h |

### C.3. Evaluation Metric and Task Category of the HELMET Benchmark.

The task category and evaluation metric of the HELMET benchmark (Yen et al., 2024b), which we use in this work, are shown in Table 10.

## D. More Details in NExtLong

### D.1. The Calculation Method for the Number of Hard Negatives k

In the Negative Document Extension stage, the meta-document targeted for extension is chunked into meta-chunks. Each meta-chunk retrieves the top-$k$ similar texts as hard negatives from the Faiss index, with the value of $k$ adaptively adjusted based on the target length $T$. Let $E$ represent the encoding rate of the model tokenizer. The total number of characters $Q$ needed for retrieval is calculated as follows:

$$Q = T \times E \times w \tag{8}$$

Here, the adjustment factor $w$ accounts for variability and ensures that a sufficient number of hard negatives are recalled for each meta-chunk. In our experiments, we set $w = 1.5$. Then, we compute the remaining characters $L$ necessary for synthesis beyond the total character length $S$ of the meta-document. This is calculated by subtracting $S$ from $Q$:

$$L = Q - S \tag{9}$$

*Table 10.* Summary of datasets and metrics in HELMET benchmark.

| Category | Dataset | Metrics | Description |
|---|---|---|---|
| Retrieval-augmented generation | Natural Questions | SubEM | Factoid question answering |
| | TriviaQA | SubEM | Trivia question answering |
| | PopQA | SubEM | Long-tail entity question answering |
| | HotpotQA | SubEM | Multi-hop question answering |
| Passage re-ranking | MS MARCO | NDCC@10 | Rerank passage for a query |
| Long-document QA | NarrativeQA | ROUGE F1 | Book and movie script QA |
| | ∞BENCH QA | ROUGE F1 | Novel QA with entity replacement |
| | ∞BENCH MC | Accuracy | Novel multiple-choice QA with entity replacement |
| Many-shot in-context learning | TREC Coarse | Accuracy | Question type classification, 6 labels |
| | TREC Fine | Accuracy | Question type classification, 50 labels |
| | NLU | Accuracy | Task intent classification, 68 labels |
| | BANKING77 | Accuracy | Banking intent classification, 77 labels |
| | CLINC150 | Accuracy | Intent classification, 151 labels |
| Synthetic recall | JSON KV | SubEM | Retrieve a key in JSON dictionary |
| | RULER MK Needle | SubEM | Retrieve the needle (a number) within noisy needles |
| | RULER MK UUID | SubEM | Retrieve the needle (a UUID) within noisy needles |
| | RULER MV | SubEM | Retrieve multiple values for one needle (key) |

The meta-document is divided into $p$ meta-chunks. The number of characters $P$ required for retrieval from each meta-chunk is distributed evenly across all $p$ meta-chunks:

$$P = \frac{L}{p} \tag{10}$$

The number of hard negatives $k$ that need to be retrieved for each meta-chunk can be calculated as follows:

$$k = \frac{P}{s} \tag{11}$$

Substituting $P$ into this equation gives:

$$k = \frac{L}{p \times s} = \frac{Q - S}{p \times s} = \frac{T \times E \times w - S}{p \times s} \tag{12}$$

This formulation ensures that the number of hard negatives $k$ is proportional to the chunking granularity $s$ and the target length $T$. Additionally, to enhance content diversity, we ensure that the same hard negative is not repeatedly used across different meta-chunks.

### D.2. Pseudocode of NExtLong

We present the complete process of constructing the NExtLong dataset using pseudocode in Algorithm 1.

---

**Algorithm 1** Negative Document Extension (NExtLong)

---

**Input:** Training corpus $\mathcal{D} = \{d_1, d_2, \ldots, d_N\}$, chunking granularity $s$, number of retrieved hard negatives $k$, target long length $L_{\text{target}}$, Faiss index *Faiss*
**Output:** Synthesized long documents for long-dependence modeling

1:  **Initialize** an empty list $\mathcal{T}$ for storing synthesized long documents

2:  **function** DOCUMENT_CHUNKING$(r, s)$
3:     **Split** $r$ into paragraphs $r = \{r_1, r_2, \ldots, r_P\}$ by newline characters
4:     Initialize an empty list chunks $= []$
5:     Initialize an empty buffer buffer $= []$ with length counter $\ell = 0$
6:     **for each** paragraph $r_i$ in $r$ **do**
7:       **if** $\ell + \text{Length}(r_i) \leq s$ **then**
8:         buffer $\leftarrow$ buffer $\cup\, r_i$
9:         $\ell \leftarrow \ell + \text{Length}(r_i)$
10:       **else**
11:         chunks $\leftarrow$ chunks $\cup\, \{\text{buffer}\}$
12:         buffer $\leftarrow r_i$
13:         $\ell \leftarrow \text{Length}(r_i)$
14:       **end if**
15:     **end for**
16:     **if** buffer $\neq \varnothing$ **then**
17:       chunks $\leftarrow$ chunks $\cup\, \{\text{buffer}\}$
18:     **end if**
19:     **return** chunks
20: **end function**

21: **function** NEGATIVE_MINING$(m_i, k, \text{\textit{Faiss}})$
22:     $n_{i_1}, n_{i_2}, \ldots, n_{i_k} \leftarrow$ Top-$k$ similar chunks from *Faiss* to $m_i$
23:     **return** $[\, m_i, n_{i_1}, \ldots, n_{i_k}\,]$
24: **end function**

25: **procedure** NEXTLONG$(\mathcal{D}, s, k, L_{\text{target}}, \text{\textit{Faiss}})$
26:     **Build Faiss index** by segmenting each $d_j \in \mathcal{D}$ into chunks
27:     Insert the embeddings of all chunks into *Faiss*
28:     **for each** document $r \in \mathcal{D}$ **do**
29:       $\{m_1, m_2, \ldots, m_p\} \leftarrow$ DOCUMENT_CHUNKING$(r, s)$
30:       **for** $i \leftarrow 1$ to $p$ **do**
31:         $l_i \leftarrow$ NEGATIVE_MINING$(m_i, k, \text{\textit{Faiss}})$
32:       **end for**
33:       $t \leftarrow [\, l_1, l_2, \ldots, l_p\,]$                     $\triangleright$ Concatenate into a single long document
34:       **if** Length(t) $\geq L_{\text{target}}$ **then**
35:         $\mathcal{T} \leftarrow \mathcal{T} \cup \{t\}$
36:       **end if**
37:     **end for**
38:     **return** $\mathcal{T}$                                    $\triangleright$ Synthesized long documents for training
39: **end procedure**

---

