# OpenReview forum: "NExtLong: Toward Effective Long-Context Training without Long Documents"
_ICML.cc/2025/Conference — ICML 2025 poster_

### Official Review · Reviewer_HH7d · 2025-03-05

**Overall Recommendation:** 4

**Summary:**

This paper introduces NExtLong, an effective framework that improves long-context modeling in LLMs through negative document extension. It first divides a document into meta-chunks and then inserts hard negative distractors to force LLMs learn the dependency between long documents. Experimental results have illustrated the efficiency of NExtLong.

## update after rebuttal
The author's rebuttal addressed my previous concerns and thus I have updated my score correspondingly.

**Claims And Evidence:**

The effectiveness of the proposed method NExtLong is proved by the experimental results.

**Essential References Not Discussed:**

N/A

**Experimental Designs Or Analyses:**

Experimental settings are good except Table1 (please see the weakness section)

**Methods And Evaluation Criteria:**

Yes.

**Other Comments Or Suggestions:**

N/A

**Other Strengths And Weaknesses:**

Weaknesses:

1. Regarding Table 1, it seems that NExtLong is using extra negatives when pre-training on the two datasets compared to other baselines, meaning that it may use more training examples, which makes the comparison not fair.

I am happy to raise my score if the authors could answer my question clearly (or point out my misunderstanding)

**Questions For Authors:**

Please see the weakness section.

**Relation To Broader Scientific Literature:**

The proposed method is novel and effective.

**Theoretical Claims:**

No issues.

---

> ### Author Rebuttal · Authors · 2025-03-29
>
> We sincerely appreciate your response! Your detailed and insightful feedback plays a crucial role in improving our article. The following text further clarifies some questions.
>
> ---
>
> **Q1: "Regarding Table 1, it seems that NExtLong is using extra negatives when pre-training on the two datasets compared to other baselines, meaning that it may use more training examples, which makes the comparison not fair."**
>
>
> **A1:** Thanks for your question. We appreciate the opportunity to clarify this point.
>
> We would like to clarify that **NExtLong does NOT use any additional training example (lines 251-256)**. The negatives utilized by NExtLong are also drawn from the two datasets (lines 208-212). Specifically, we construct a FAISS index from these datasets, and NExtLong retrieves hard negatives from the FAISS index to synthesize long-context data. The FAISS index used by the baseline methods is also built from the same datasets; **the only difference between those methods and ours is how they rearrange documents into long-context data.**
>
>
> **Importantly, the total number of training examples and the experimental settings are identical across all methods, ensuring a fair comparison.** This is explicitly stated in the following sections of our paper:
>
> - Lines 208-212: "*Various methods, including NExtLong and baseline approaches, are employed to synthesize target-length samples concatenated from these short documents.*"
> - Lines 251-256: "*...The same training configuration is applied to all methods for a fair comparison.*"
> - Lines 263-272 provide further details on how baseline methods synthesize long-context data.
>
> We apologize for any confusion and hope this clarification addresses your concern. Please let us know if any further details would be helpful.

---

### Official Review · Reviewer_hZK6 · 2025-03-12

**Overall Recommendation:** 3

**Summary:**

The paper introduces the NExtLong framework, which aims to alleviate issues arising from the scarcity of high-quality long documents in long-context training. Traditional methods that concatenate shorter documents do not effectively capture necessary long-range dependencies, leading to problems with coherence and relevance. NExtLong addresses these challenges by decomposing long documents into meta-chunks and incorporating hard negative distractors from pre-training corpora. This method increases the difficulty of the training process, encouraging the model to differentiate between relevant long-range dependencies and distracting information, thereby improving its overall modeling capabilities. Experimental results indicate that NExtLong outperforms existing long-context synthesis methods and state-of-the-art models, yielding an average performance improvement of 7.33% on key benchmarks such as HELMET and RULER.

**Claims And Evidence:**

This work claims that the proposed data cooking method is effective for long-context training. This main claim is supported by clear and convincing evidence. In the experimental section, the authors compare the proposed method with various alternatives and demonstrate the effectiveness of hard negatives through ablation experiments (section 5.4).

**Essential References Not Discussed:**

Zhao, L., Wei, T., Zeng, L., Cheng, C., Yang, L., Cheng, P., ... & Zhou, Y. (2024). Longskywork: A training recipe for efficiently extending context length in large language models. arXiv preprint arXiv:2406.00605.

I would like to suggest to include the above work in the related section. It is the first work try to use interleaved chunk data to improve long-context performance.

**Experimental Designs Or Analyses:**

The experimental design is standard and sound.

**Methods And Evaluation Criteria:**

The proposed method is evaluated on the two most important long-context benchmarks: RULER and HELMET.

**Other Comments Or Suggestions:**

1. For Table 1, I suggest that the authors provide the results averaged over 32K, 64K, and 128K in the appendix, following the same evaluation protocol as in ProLong.

**Other Strengths And Weaknesses:**

In general, this work is solid, with well-presented experimental results.

Weaknesses:
1. Limited Novelty:
   The novelty is limited given the existence of previous works [1] and [2]. Compared with [1], this work omits training strategies like Knot Tokens and instead introduces a negative sample data preparation strategy. Moreover, [2] was actually the first to use interleaved chunks to achieve improved performance. The authors should, at a minimum, include [2] in the related work.

2. Compatibility with Modern Techniques:
   The proposed data preparation method is not compatible with the commonly used intra-document attention approach for long-context training. After Llama3 demonstrated the effectiveness of intra-document attention, in terms of both performance improvement and reduced memory requirement, the Qwen series and recent research works [3, 4] have adopted this approach. While the proposed method can improve the older full attention mechanism, it may not compare favorably with emerging methods.

----

[1] Tian, J., Zheng, D., Cheng, Y., Wang, R., Zhang, C., & Zhang, D. (2024). Untie the knots: An efficient data augmentation strategy for long-context pre-training in language models. *arXiv preprint arXiv:2409.04774.

[2] Zhao, L., Wei, T., Zeng, L., Cheng, C., Yang, L., Cheng, P., ... & Zhou, Y. (2024). Longskywork: A training recipe for efficiently extending context length in large language models. *arXiv preprint arXiv:2406.00605.

[3] Wang, H., Liu, Q., Du, C., Zhu, T., Du, C., Kawaguchi, K., & Pang, T. (2024). When Precision Meets Position: BFloat16 Breaks Down RoPE in Long-Context Training. *arXiv preprint arXiv:2411.13476.

[4] Gao, T., Wettig, A., Yen, H., & Chen, D. (2024). How to train long-context language models (effectively). arXiv preprint arXiv:2410.02660.

**Questions For Authors:**

1. In Table 2, the results are averaged over 8K, 16K, 32K, 64K, and 128K. Does Table 5 follow the same procedure? I suggest that the authors provide more detailed explanations of the experimental setup.

**Relation To Broader Scientific Literature:**

The key findings regarding the data composition strategy have the potential to enhance the long-context training methods used in current LLMs.

**Theoretical Claims:**

N/A

---

> ### Author Rebuttal · Authors · 2025-03-29
>
> We sincerely appreciate your response! Your detailed and insightful feedback plays a crucial role in improving our article. The following text further clarifies some questions.
>
> ---
>
> **Q1: "I would like to suggest to include the above work in the related section. It is the first work try to use interleaved chunk data to improve long-context performance."**
>
> **A1:** We sincerely appreciate your valuable suggestion. We acknowledge the significance of the mentioned work as the first to explore the use of interleaved chunk data for improving long-context performance. **We will incorporate this important reference in the revised version** to ensure a more comprehensive discussion in the related work section. Thank you for bringing this to our attention.
>
> **Q2: "Limited Novelty: The novelty is limited given the existence of previous works [1] and [2]."**
>
> **A2:** We would like to emphasize that **our contribution is not using interleaved chunks**. As we explicitly state in lines 55–57:  *"An intuitive approach to building documents with long-range dependencies is to insert additional text between dependent segments.[1]"*  Additionally, we discuss the limitations of those methods in lines 62–65, noting that **they do not fully capture the real-world challenge of extracting long-range dependencies amid extensive distracting information.**
>
> We replace hard negatives with random chunks to implement a baseline (RandomD) similar to [1][2], and Figure 7 shows that it underperforms NExtLong. The results indicate that our method better reflects real-world scenarios where models must extract relevant information despite interference, thereby fostering more robust long-range dependency learning (As mentioned in lines 83-88).
>
> **Q3: "Compatibility with Modern Techniques: The proposed data preparation method is not compatible with the commonly used intra-document attention approach for long-context training."**
>
> **A3:** We would like to clarify that **our method is indeed compatible with intra-document attention**. In fact, **Llama-3-8B-NExtLong-512K-Base is trained using the intra-document attention approach and achieves strong results (Table 2)**. Specifically, we construct 512K and 64K datasets (lines 326–329) and apply intra-document attention to the 64K subsets, enabling their concatenation into 512K context length. This strategy aligns with ProLong[4] (lines 318–320), which uses a 64K dataset to train 512K models. Our results in Table 2 demonstrate that NExtLong performs better under this setup.
>
> **Q4: "For Table 1, I suggest that the authors provide the results averaged over 32K, 64K, and 128K in the appendix, following the same evaluation protocol as in ProLong."**
>
> **A4:** Thank you for the suggestion. We will incorporate the results averaged over 32K, 64K, and 128K in the appendix. We report the averaged results in the table below, and NExtLong maintains a performance advantage across sequence lengths beyond 16K, further supporting the effectiveness of our approach.
>
> | Model                              | Avg.  | Recall | RAG   | ICL   | Re-rank | LongQA | RULER |
> |------------------------------------|-------|--------|-------|-------|---------|--------|-------|
> | Llama-3-8B-ProLong-512K-Base       | 60.34 | 82.38  | 60.10 | 84.07 | 24.77   | 33.44  | 77.26 |
> | Llama-3-8B-NExtLong-512K-Base      | **64.71** | **87.96**  | **62.42** | **88.67** | **25.81**   | **41.27**  | **82.14** |
>
> **Q5: "In Table 2, the results are averaged over 8K, 16K, 32K, 64K, and 128K. Does Table 5 follow the same procedure? I suggest that the authors provide more detailed explanations of the experimental setup."**
>
> **A5:** Yes, **Table 5 follows the same reporting procedure**. Specifically, the "Head" strategy in Table 5 corresponds to the NExtLong results in Table 1.
>
> We sincerely apologize for any confusion caused by the insufficient explanation of the experimental setup. We will include more details in the revised version. Please let us know if any further information would be helpful.
>
> ---
>
> [1] Tian, J., Zheng, D., Cheng, Y., Wang, R., Zhang, C., & Zhang, D. (2024). Untie the knots: An efficient data augmentation strategy for long-context pre-training in language models. *arXiv preprint arXiv:2409.04774.
>
> [2] Zhao, L., Wei, T., Zeng, L., Cheng, C., Yang, L., Cheng, P., ... & Zhou, Y. (2024). Longskywork: A training recipe for efficiently extending context length in large language models. *arXiv preprint arXiv:2406.00605.
>
> [3] Wang, H., Liu, Q., Du, C., Zhu, T., Du, C., Kawaguchi, K., & Pang, T. (2024). When Precision Meets Position: BFloat16 Breaks Down RoPE in Long-Context Training. *arXiv preprint arXiv:2411.13476.
>
> [4] Gao, T., Wettig, A., Yen, H., & Chen, D. (2024). How to train long-context language models (effectively). arXiv preprint arXiv:2410.02660.

---

> > ### Comment · Reviewer_hZK6 · 2025-04-05
> >
> > Thanks for your reply.
> >
> > Could the authors explain more about how the proposed method compatible with intra-document attention? The inter-doc masking of it will block the information flow between documents.

---

> > > ### Author Response · Authors · 2025-04-06
> > >
> > > Previous studies [1, 2, 3] employ intra-document attention on training samples formed by concatenating multiple documents. For instance, ProLong [1] uses 512K-length and 64K-length documents when training the 512K-length model (refer to Table 9 in the ProLong paper). These 64K-length documents are concatenated into 512K-length training samples — each sample contains eight 64K-length documents — and intra-document attention is applied to prevent interference among these documents.
> > >
> > > Given that mixing documents of different lengths during training is widely adopted in prior work [1, 4, 5], we follow the ProLong approach by synthesizing both 512K-length and 64K-length documents for training the 512K NExtLong model (lines 326–329). For synthetic 512K-length documents, we apply the full attention mechanism, as each document constitutes an entire training sample. For synthetic 64K-length documents, we concatenate them into 512K-length training samples (each sample contains eight 64K-length documents) and employ intra-document attention to restrict information flow within each 64K-length document.
> > >
> > > In addition, as shown in Table 1, our 128K NExtLong model trained on 128K synthetic documents (without intra-document attention or advanced techniques such as train-long/test-short) outperforms both the ProLong model and Llama3.1 on average (62.58 vs. 60.34 and 61.07, respectively). For a fair comparison (lines 313-320), we combine advanced strategies and achieve better results with our 512K NExtLong model (65.76 on average).
> > >
> > > Our experimental results demonstrate that NExtLong is capable of synthesizing documents of arbitrary lengths (in this work, we synthesize documents of 64K, 128K, and 512K lengths) and is adaptable to various modern techniques, ensuring both flexibility and effectiveness.
> > >
> > > We will further emphasize these experimental settings in the revised version. Thank you for your constructive feedback！
> > >
> > > ---
> > >
> > > [1] Gao, T., Wettig, A., Yen, H., & Chen, D. (2024). How to train long-context language models (effectively). arXiv preprint arXiv:2410.02660.
> > >
> > > [2] Ding, H., Wang, Z., Paolini, G., Kumar, V., Deoras, A., Roth, D., & Soatto, S. (2024). Fewer truncations improve language modeling. In Proceedings of the 41st International Conference on Machine Learning (ICML'24).
> > >
> > > [3] Meta. (2024). Introducing meta llama 3: The most capable openly available llm to date.
> > >
> > > [4] Fu Y, Panda R, Niu X, et al. Data engineering for scaling language models to 128k context. (2024). In Proceedings of the 41st International Conference on Machine Learning (ICML'24).
> > >
> > > [5] Xiong et al. (2024). Effective Long-Context Scaling of Foundation Models. In Proceedings of the 2024 Conference of the North American Chapter of the Association for Computational Linguistics (NAACL'24).

---

### Official Review · Reviewer_3FUJ · 2025-03-14

**Overall Recommendation:** 4

**Summary:**

The paper introduces NExtLong, a new framework designed to address the challenge of training LLMs with extended context windows, particularly in the face of limited availability of long documents. The key contributions of the paper are the usage of hard negative documents mining for the construction of long documents. Experimental results on RULER and HELMET show the strong performance of this paper.

**Claims And Evidence:**

The main claim is that the proposed method enhances long-context modeling by synthesizing data with hard negatives, addressing the scarcity of long documents.

Evidence: It outperforms all baselines by a large margin (e.g., +7% over Quest) on HELMET and RULER (Table 1). The superior performance on long-context ICL is especially impressive.

**Essential References Not Discussed:**

No

**Experimental Designs Or Analyses:**

Comparisons: Tested against KNN, Quest, and SOTA models (Llama-3 and other open models, with GPT-4o, Gemini-1.5-Pro and Claude).
Ablations: Analyzed chunking granularity (Figure 6), negative selection strategies (Figure 7), and dataset combinations (Table 7).

**Methods And Evaluation Criteria:**

Method: chunk documents into meta-chunks, retrieve hard negatives via FAISS, interleave them, and train with next-token prediction loss.
Evaluation: Two commonly used long-context benchmark, including HELMET and RULER (several subtasks), measuring recall, RAG, ICL, re-ranking, LongQA, and synthetic tasks. Metrics include Accuracy and ROUGE F1.

**Other Comments Or Suggestions:**

N/A

**Other Strengths And Weaknesses:**

Strengths: neat and good use of hard negatives, comprehensive evaluations, minimal performance drop on short-text tasks.
Weaknesses: Dependency on FAISS retrieval quality, computational cost for indexing, not comparing against models using realistic long documents.

**Questions For Authors:**

Although I understand it is not the main point of this paper, but I am curious about if the proposed method can be applied to long documents to further extend the effective context length of the model? For example, using a dataset which contains more documents > 8K length, and apply the proposed method on it to get a much better long-context performance on the evaluation benchmark.

**Relation To Broader Scientific Literature:**

Builds on contrastive learning (hard negatives) and long-context methods (e.g., ProLong, Quest). Addresses document scarcity, a key challenge in long-context model training.

**Theoretical Claims:**

No, no theoretical claim in this paper.

---

> ### Author Rebuttal · Authors · 2025-03-29
>
> We sincerely appreciate your response! Your detailed and insightful feedback plays a crucial role in improving our article. The following text further clarifies some questions.
>
> ---
>
> **Q1: Dependency on FAISS retrieval quality**
>
> **A1:** The quality of FAISS retrieval depends significantly on the effectiveness of the embedding model. As embedding models advance, retrieval performance is expected to improve accordingly. Given ongoing advancements in this area [1][2], we anticipate these advancements will further enhance the performance of NExtLong and plan to explore the impact of more advanced embedding models in future work.
>
> ---
>
> [1] Choi, Chanyeol, et al. Linq-Embed-Mistral Technical Report. arXiv preprint arXiv:2412.03223 (2024).
>
> [2] Wang, Liang, et al. Multilingual e5 text embeddings: A technical report. arXiv preprint arXiv:2402.05672 (2024).
>
>
> **Q2: "computational cost for indexing"**
>
> **A2:** The computational cost of indexing is only incurred during the generation of synthetic training data and does not impact the inference phase of the model. Since this is a one-time cost, the resulting performance improvements provide lasting benefits for downstream tasks. Given these long-term advantages, the computational expenditure is justified and contributes to overall performance.
>
> **Q3: "not comparing against models using realistic long documents."**
>
> **A3:** We would like to clarify that **some of the models compared in Table 2 are indeed trained with realistic long documents**. For example, ProLong [1] uses realistic long documents (as mentioned in lines 323-325) and extensively explores how to utilize them effectively.
>
> To ensure a fair comparison, Table 2 directly compares NExtLong with ProLong. It shows that Llama-3-8B-NExtLong-512K-Base outperforms Llama-3-8B-ProLong-512K-Base, further validating the effectiveness of NExtLong (lines 319-322).
>
> ---
>
> [1] Gao, Tianyu, et al. How to train long-context language models (effectively). arXiv preprint arXiv:2410.02660 (2024).
>
> **Q4: "Although I understand it is not the main point of this paper, but I am curious about if the proposed method can be applied to long documents to further extend the effective context length of the model? ..."**
>
> **A4:** Based on the hypothesis that shorter original documents pose more significant challenges for a fixed target length and given that training a long-context model is resource-intensive, we prioritized creating a more challenging experimental setting within our limited training resources. Our current experiments deliberately use relatively short documents, creating an effective 64x increase in sequence length compared to the original documents (as mentioned in lines 651–652).
>
> Despite this challenging setting, NExtLong still achieved strong performance, **leading us to believe that when the original documents are longer, they can be merged into even longer sequences while maintaining promising results**. We are currently collecting longer documents and will systematically investigate the impact of documents exceeding 8K tokens in future work.
>
> Thank you for your insightful question!

---

### Official Review · Reviewer_syXt · 2025-03-21

**Overall Recommendation:** 4

**Summary:**

Most LLMs require long documents to handle long-context processing, but in practice, high-quality long documents are scarce. Existing methods typically concatenate short documents either randomly or based on similarity, which is not effective for learning long-range dependencies. This paper proposes splitting documents into multiple meta-chunks and inserting hard negatives—text segments that are semantically similar but unrelated to the actual context—between them. This encourages the model to distinguish between true context and misleading context, thereby enhancing its ability to learn long-range dependencies.

**Claims And Evidence:**

Yes, it's supported by the experiments section.

**Essential References Not Discussed:**

N/A

**Experimental Designs Or Analyses:**

I checked the validity of the experimental designs. It would be better to explain why improving the long-context ability of LLMs can decrease the performance on short text benchmarks.

**Methods And Evaluation Criteria:**

The proposed methods and evaluation criteria seem make sense for the problem.

**Other Comments Or Suggestions:**

N/A

**Other Strengths And Weaknesses:**

**Strengths**
1. Writing is clear.
2. The motivation behind the method is clear.
3. The efficacy of the method is demonstrated through the experiments on HELMET and RULER benchmark.

**Weaknesses**
1. Compared to prior methods, the novelty of this approach feels somewhat limited.
2. It would be helpful to more clearly highlight which aspects of the previous methods made them suboptimal.

**Questions For Authors:**

Please see above.

**Relation To Broader Scientific Literature:**

Improving the ability of LLMs to handle long context is important, and this paper addresses how to effectively construct synthesized long documents.

**Theoretical Claims:**

N/A

---

> ### Author Rebuttal · Authors · 2025-03-29
>
> We sincerely appreciate your response! Your detailed and insightful feedback plays a crucial role in improving our article. The following text further clarifies some questions.
>
> ---
>
> **Q1: "It would be better to explain why improving the long-context ability of LLMs can decrease the performance on short text benchmarks."**
>
> **A1:**
> Previous long-context extension approaches [1] often degrade short-context performance because they rely exclusively on non-synthetic long documents, which are scarce across most domains (as noted in lines 35–37). This view also aligns with [2], which states that "*a data mixture that keeps the domain mixture ratio the same as the pretraining mixture...this is the primary reason our solution improves long context tasks while maintaining short context performance.*"
>
> Quest [3] demonstrates that using a long-text dataset synthesized from short texts with good domain diversity does not decline short-text performance. Following Quest, we also use short-text data sources with sufficient diversity, such as Cosmopedia V2, which covers over 34,000 topics. As a result, Table 5 demonstrates that NExtLong does not degrade short-text performance on average.
>
> Thanks for your valuable suggestion! We will elaborate on this point in the revised version.
>
> **Q2: "Compared to prior methods, the novelty of this approach feels somewhat limited. It would be helpful to more clearly highlight which aspects of the previous methods made them suboptimal."**
>
> **A2:** As mentioned in lines 51-53: "*those methods typically concatenate short documents based on random or similarity-based rankings, lacking a clear mechanism for capturing long-range dependencies.*" Previous studies (e.g., Quest [3]) synthesize long-context data by concatenating similar documents sequentially, allowing models to perform the next token prediction without relying on preceding documents. That concatenation strategy weakens the learning of long-range dependencies, and Table 1 shows that those methods achieve only marginal improvements compared to the Standard approach.
>
> While approaches like [4][5] increase dependency length using interleaved chunk data, they do not capture the real-world challenge of extracting long-range dependencies amid extensive distracting information. To verify this, we replace hard negatives with random chunks to implement a baseline (RandomD) similar to [4][5], and Figure 7 shows that it underperforms NExtLong.
>
> In contrast, as mentioned in lines 83-88: "*By inserting these distractors between originally dependent meta-chunks, NExtLong not only increases the distance between dependent chunks—effectively transforming the dependencies into long-range ones—but also introduces distracting noise.*" **This design better reflects real-world scenarios where models must extract relevant information despite interference, thereby fostering more robust long-range dependency learning.**
>
> We will provide a more detailed comparative analysis in the revised version. Thanks for your constructive suggestion!
>
> ---
>
> [1] Chen Y, Qian S, Tang H, et al. Longlora: Efficient fine-tuning of long-context large language models. The Twelfth International Conference on Learning Representations (ICLR'24).
>
> [2] Fu Y, Panda R, Niu X, et al. Data engineering for scaling language models to 128k context. In Proceedings of the 41st International Conference on Machine Learning (ICML'24).
>
> [3] Gao C, Wu X, Fu Q, et al. Quest: Query-centric data synthesis approach for long-context scaling of large language model. The Thirteenth International Conference on Learning Representations (ICLR'25).
>
> [4] Tian, Junfeng, et al. Untie the knots: An efficient data augmentation strategy for long-context pre-training in language models. arXiv preprint arXiv:2409.04774 (2024).
>
> [5] Zhao, L., Wei, T., et al. Longskywork: A training recipe for efficiently extending context length in large language models. arXiv preprint arXiv:2406.00605 (2024).

---

### Decision · Program_Chairs · 2025-05-01

**Decision:**

Accept (poster)

**Comment:**

This paper tackles a critical bottleneck in long‑context LLM pretraining—namely, the scarcity of high‑quality, naturally long documents—by proposing NExtLong, a synthetic data framework that interleaves “hard negative” distractors between contiguous meta‑chunks. By forcing the model to distinguish true long‑range dependencies from semantically similar but irrelevant text, NExtLong significantly boosts long‑context understanding without degrading short‑text performance. Empirical results on HELMET and RULER benchmarks demonstrate consistent gains (≈7 % average improvement) over strong baselines—including Quest, ProLong, and Llama‑3 variants—across multiple context lengths (8 K–512 K).

All reviewers elevated or maintained their scores after rebuttal. The paper addresses an important problem, presents a clear, well‑motivated method, and achieves substantial empirical gains. While the core idea bears similarity to previous work (e.g., LongSkywork), NExtLong’s use of retrieval‑based hard negatives and its systematic study across extremely long contexts (up to 512 K) represent a meaningful advance. The requested clarifications and added comparisons can be accommodated in the camera‑ready.